# Roles of Mesenchymal Cells in the Lung: From Lung Development to Chronic Obstructive Pulmonary Disease

**DOI:** 10.3390/cells10123467

**Published:** 2021-12-09

**Authors:** Amel Nasri, Florent Foisset, Engi Ahmed, Zakaria Lahmar, Isabelle Vachier, Christian Jorgensen, Said Assou, Arnaud Bourdin, John De Vos

**Affiliations:** 1Institute for Regenerative Medicine and Biotherapy, Université de Montpellier, INSERM, Centre Hospitalier Universitaire de Montpellier, 34000 Montpellier, France; amel.nasri@inserm.fr (A.N.); florent.foisset@inserm.fr (F.F.); christian.jorgensen@inserm.fr (C.J.); said.assou@inserm.fr (S.A.); 2Department of Respiratory Diseases, Université de Montpellier, INSERM, Centre Hospitalier Universitaire de Montpellier, 34090 Montpellier, France; noussa31@gmail.com (E.A.); zakaria.lahmar@gmail.com (Z.L.); isabelle.vachier@medbiomed.fr (I.V.); a-bourdin@chu-montpellier.fr (A.B.); 3PhyMedExp, Université de Montpellier, INSERM, Centre Hospitalier Universitaire de Montpellier, 34295 Montpellier, France; 4Department of Cell and Tissue Engineering, Université de Montpellier, Centre Hospitalier Universitaire de Montpellier, 34000 Montpellier, France

**Keywords:** mesenchyme, lung, development, COPD

## Abstract

Mesenchymal cells are an essential cell type because of their role in tissue support, their multilineage differentiation capacities and their potential clinical applications. They play a crucial role during lung development by interacting with airway epithelium, and also during lung regeneration and remodeling after injury. However, much less is known about their function in lung disease. In this review, we discuss the origins of mesenchymal cells during lung development, their crosstalk with the epithelium, and their role in lung diseases, particularly in chronic obstructive pulmonary disease.

## 1. Introduction

The lung is a complex organ that carries out the vital task of blood oxygenation. To offer the surface required for this process, the lung gas exchange units (i.e., the alveoli) and the corresponding airways have been multiplied through iterative, fractal branching. This comes at a cost: the direct contact with the external environment of a surface larger than the skin. This contact surface is estimated to be bigger than 100 square meters. The bronchial epithelium is composed of polarized cells attached to a basement membrane and is closely interconnected through junctions, such as adherent junctions, tight junctions, gap junctions, desmosomes and hemi-desmosomes. The bronchial epithelium is composed of different cell types, including ciliated cells, club cells and goblet cells. It functions as a protective barrier and maintains the tissue homeostasis [1,2,3]. The main protection of the epithelium against external aggression (dusts, airborne particulate matters, noxious compounds, and microbes) is the mucociliary clearance system [4]. Goblet cells secrete mucin glycoproteins to form the mucus that covers bronchioles with a gel layer to trap the inhaled physical matter. Then, cilia on the ciliated cell surface beat to remove these pollutants from the airways.

Like all epithelia, the bronchial epithelium relies on the lung mesenchyme for physical support, nutrient supply [5,6,7,8], and key differentiation cues during development. The composition of the lung mesenchyme is still not fully known, but it comprises a large array of cells: endothelial cells, lymphatic cells, pericytes, fibroblasts, mesenchymal stromal cells, smooth muscle cells, and myofibroblasts [9,10,11,12,13]. Mesenchymal stromal cells and fibroblasts are morphologically indistinguishable, and they are differentiated mainly using functional assays [14]. In this review, these different cell types are collectively called “mesenchymal cells”. They have critical supporting roles and display specific features, such as migration and invasion [15]. Gene expression analysis and more recently single-cell mRNA sequencing have been useful to explore the mesenchymal cell diversity. For instance, their transcriptomic signature varies according to their localization in the body [11,14,16,17,18,19]. Moreover, a comprehensive catalogue of all mesenchymal cells has been proposed for human [20], and mouse [21] lungs. This list includes already known cell types (airway smooth muscle cells, adventitial fibroblasts, lipofibroblasts, mesothelial cells, myofibroblasts, pericytes, vascular smooth muscle cells), and also novel cell types identified on the basis of their gene expression profile, such as fibromyocytes and alveolar fibroblasts. All these mesenchymal cell types harbor a distinct gene expression signature (Figure 1 shows some of these cell types and their gene expression signature). Discrepancies among studies may reflect differences linked to the exact tissue localization or the species investigated (proximal versus distal pattern species, human versus mouse, etc.), and suggest that the different mesenchymal cell types could represent different steps in a continuous process of differentiation that varies also between different cell types but also between healthy and diseased lung. As the lung mesenchymal compartment contributes to lung homeostasis and repair after injury [22,23], mesenchymal cells are implicated in many lung diseases, particularly asthma, idiopathic pulmonary fibrosis, and chronic obstructive pulmonary disease (COPD). In this review, we highlight the importance of the crosstalk between epithelium and mesenchyme during lung development, adult life, and diseases, with a focus on COPD.

## 2. Origin of Pulmonary Mesenchymal Cells

At the bilaminar disc stage in the second week of human development, gastrulation starts with the formation of the primitive streak [30]. Cells forming the epiblast undergo epithelial to mesenchymal transition (EMT) and migrate through the primitive streak to form the endoderm and mesoderm cell layers (Figure 2A) [31]. Specifically, the first wave of cells integrates the hypoblast layer and forms the endoderm [30], from which the lung will derive. A second group of cells migrates between the epiblast and endoderm layers and constitutes the mesoderm layer that will give rise to a large variety of tissues, such as skeletal muscle, bone, cartilage, and many mesenchymal cell types (e.g., fibroblasts, chondroblasts, osteoblasts, blood cells). The mesoderm is a major contributor to trunk and limb stromal cells, but neural crest cells also contribute to mesenchymal cell lineages [32,33], although their specific role in lung mesenchyme is still poorly defined.

## 3. Mesenchymal Cells during Lung Organogenesis

At the beginning of the fourth week of development, the anterior foregut endoderm develops at the cranial extremity after the cephalocaudal folding of the embryo. The foregut produces a ventral evagination that leads to lung bud development. These buds are surrounded by the splanchnopleuric mesoderm that is part of the lateral mesoderm and will contribute to lung vascularization, cartilage, muscles and conjunctive tissue (Figure 2B). In addition, the embryonic lateral splanchnic mesoderm generates mesothelial cells that form a thin layer of squamous-like cells lining the visceral pleura (mesothelium) [34,35,36]. Then, at day 26, the lung buds divide into right and left primitive bronchial buds, which are the precursors of the two lungs (Figure 2B).

During the pseudo-glandular stage, a second division at week 5 of development leads to the formation of the future pulmonary lobes by creating three secondary bronchial buds on the right and two on the left side. Each lung bud and the surrounding splanchnopleuric mesoderm grow, elongate and branch until the formation of the terminal bronchioles (17th branching generation) to create the respiratory tree (Figure 2B) [37]. At this stage, the tracheobronchial tree is coated by prismatic epithelial cells, the precursors of ciliated and secretory cells. Bronchioles appears during the canalicular stage (week 16 to 25), forming the basis of the gas exchange units. This is accompanied by geometric modifications of epithelial cells that flatten and by the appearance of capillaries throughout the mesenchyme that surround the bronchioles. Finally, at the saccular stage (week 24 to 40), alveolar ducts start to form. Their formation will continue after birth, and will terminate only in adulthood [7]. Of note, the development of the gas exchange units in utero and during early childhood is critical for achieving full adult lung function [38]. Lung development is the consequence of an interweaved relationship between embryonic lung epithelial and mesenchymal cells, through direct interactions and also indirectly via the secretion of extracellular matrix (ECM) components and growth factors [39,40,41]. Moreover, the mesothelium plays an important role during lung development [36,42,43], partly by secreting fibroblast growth factor (FGF) 9 [35,44,45].

Fetal airway smooth muscle (ASM) development begins early during gestation (from week 5–6 in human airways) [46]. Fetal ASM surrounds the airways and guides lung development and branching. ASM cells spontaneously contract early in fetal life, with proximal to distal peristaltic-like contractions that displace the amniotic liquid along the lumen [47,48]. At the pseudo-glandular and canalicular stages in pigs [48], the mechanical distention and stretching of the developing lungs, produced by ASM contractions, influence lung growth via mechanotransduction, through the pressure exerted across the airway wall and the surrounding parenchyma. Furthermore, the transmural pressure regulates the rate of airway epithelial bud branching [49].

The next part of this review will focus on some of the most relevant factors implicated in the epithelium–mesenchyme interactions during lung development.

### 3.1. Peptide Growth Factors

#### 3.1.1. Fibroblast Growth Factors

The large FGF family plays an important role in the regulation of cell differentiation, proliferation and development, including lung branching [50]. Several studies have identified FGFs implicated in the bidirectional signaling between epithelium and mesenchyme during lung development [44,50,51]. For example, FGF10 is expressed in the distal submesothelial mesenchyme and activates FGF receptor 2b (FGFR2b) in the adjacent epithelial cells (Figure 3) to induce lung budding, epithelial cell expansion and migration, and ECM organization [52,53,54]. Indeed, in *Fgf*10^−/−^ mice, the lung does not develop below the trachea [55,56]. Moreover, Bellusci et al. have identified, in mice, a subtype of Axin2+/FGF10+-resident mesenchymal alveolar niche cells that are close to alveolar type 2 (AT2) stem cells and that control the proliferation and differentiation of AT2 cells [57].

FGF9 is another FGF family member that is expressed in the mesothelium from the pseudo-glandular stage (week 5 to 17) onwards, regulating the local activation of Wnt signaling to promote mesenchymal cell proliferation [35,44] (Figure 3). FGF9 also plays a critical role in lung development, as indicated by the finding that *Fgf*9^−/−^ mice die at the neonatal stage due to lung hypoplasia caused by very reduced mesenchymal cell expansion [58].

#### 3.1.2. Bone Morphogenic Protein 4 (BMP4)

BMP4 belongs to the transforming growth factor superfamily. *Bmp*4^−/−^ mice die early during development, mainly due to an absence of mesoderm differentiation [59]. During lung development, BMP4 expression is detected in the distal epithelium buds and in the adjacent mesenchyme already at the pseudo-glandular stage (week 5 to 17) [60,61]. Conversely, BMP type I receptor (BMPR1) is expressed in both epithelium and mesenchyme (Figure 2B) [62]. In association with sonic hedgehog (SHH), BMP4 antagonizes FGF10 that is expressed in the surrounding mesenchyme [63]. Conditional knock-out of *Bmpr* 1*a*, the gene encoding the BMP4 receptor in the epithelium, leads to abnormal lung development with reduced cell proliferation, increased apoptosis and abnormal lung morphogenesis. This indicates that BMP4 plays important roles in lung development [64]. Moreover [65], in cultured mouse embryonic lung, reduction of gremlin expression, a BMP4 antagonist, using antisense oligonucleotides promotes epithelial cell proliferation and abnormal lung formation/function. Finally, BMP4 overexpression in the distal bud tips leads to lung hypoplasia, reduction of AT2 cells, and enlargement of the terminal buds [62]. However, the exact role of this signaling pathway during lung development remains debated, because mathematical models to mimic the FGF10-SHH interaction accurately model bronchial branching independently of BMP4 expression [66,67].

#### 3.1.3. Sonic Hedgehog

SHH also is part of a key developmental signaling pathway. It is implicated in central nervous system patterning, and limb, digit and lung development [68]. In *Shh*^−/−^ mice, a single lobe, lung hypoplasia with absence of left and right asymmetry, enhanced cell death and decreased lung mesenchymal cell proliferation are observed [69]. SHH is expressed with BMP4 in the distal bud epithelium during lung development. It binds to and activates its receptor Patched 1 (PTCH1) that is localized in the adjacent mesenchyme. Patched 1 activation downregulates FGF10 expression [44,52,70]. Indeed, in *Shh*^−/−^ mice, FGF10 expression in the mesenchyme immediately adjacent to the epithelium is increased [71]. In addition, BMP4 is overexpressed and wingless-related integration site family member 2 (WNT2) is downregulated in the mesenchyme. In 2013, Peng et al. [72] identified a cardiopulmonary mesoderm progenitor population that is defined by the expression of WNT2, glioma-associated oncogene 1 (Gli1) and Islet 1, and gives rise to the lung mesenchyme and cardiac lineage in the mouse. This population is regulated by Hedgehog signaling because Gli proteins are the main transcriptional effectors of this pathway, and start to be expressed at the lung organogenesis step [72]. This suggests that SHH is broadly involved in mesenchymal signaling in the developing lung.

#### 3.1.4. Epidermal Growth Factor (EGF)

EGF and its tyrosine kinase receptor EGF-R are expressed in the epithelial and mesenchymal compartments during lung development. EGF stimulates lung branching in fetal mice [40]. In agreement, in *Egfr*^−/−^ mice, neonatal lethality is high and epithelial cell development is impaired in several organs, including the lung [73]. Importantly, lung branching is reduced and alveolarization and septation are deficient in *Egfr*^−/−^ mice [74]. Similarly, mouse lung branching can be inhibited in ex-vivo cultures using antisense oligonucleotides against EGF [40,75]. Furthermore, the interplay between retinoid acid (RA) and EGFR during fetal lung development stimulates lung branching [76].

#### 3.1.5. Retinoic Acid (RA)

RA is essential for normal embryo development, including lung development [77]. In the mouse, retinoic acid receptor (RAR) double knock-out (RARα-/-/RARβ2-/-) leads to agenesis of the left lung and hypoplasia of the right lung [78]. RA is produced by the mesenchyme surrounding the lung primordia, and also by the epithelial compartment of the proximal bronchi [79]. Importantly, a RA-transforming growth factor (TGF)-β-FGF10 interaction has been described during lung bud induction, where RA downregulates TGF-β to allow FGF10 expression [80].

#### 3.1.6. TGF-β

TGF-β, a pleiotropic growth factor and a key EMT inducer, is another modulator of epithelium–mesenchyme interactions in the developing lung [81]. Its three isoforms play a crucial role during lung organogenesis [82]. Their expression is well characterized in the mouse where TGF-β1 is expressed in the mesenchyme, TGF-β2 in the distal epithelium, and TGF-β3 in the proximal mesenchyme and pulmonary mesothelium [83,84]. In *Tgf-β*2^−/−^ mice, the distal airways are collapsed and the proximal airways are dilated [82], whereas in *Tgf*-*β*3^−/−^ mice lung development is delayed, causing their death [85]. Conversely, *Tgfr*-*β*2 ablation in mesodermal tissue results in abnormal lung branching and lung development, while its ablation in epithelial cells that produce surfactant protein C results in a decrease of alveolar type I (AT1) epithelial cells during post-natal alveolarization [86]. TGF-β also plays a role in FGF10 regulation by RA [80] as illustrated by the finding that ectopic TGF-β1 expression inhibits FGF10-induced lung morphogenesis in cultured embryonic lung endodermal explants [87].

#### 3.1.7. The Hippo Pathway

The Hippo pathway and its downstream targets Yes-associated protein (YAP) and transcriptional coactivator with PDZ-binding motif (TAZ) play major roles in tissue homeostasis, organ development, and organ size [88,89] by regulating various processes, such as cell proliferation/survival, the response to mechanical stress and cell geometry. This pathway involves a kinase cascade and adaptors that ultimately regulate YAP/TAZ activities. The name comes from the Drosophila kinase Hippo, the mammalian orthologs of which are the kinases MST1/2. Upon activation of the Hippo signaling pathway, MST1/2 phosphorylates LATS1/2, thereby activate these kinases that in turn phosphorylate YAP and TAZ. Phosphorylated YAP and TAZ are retained in the cytoplasm or degraded by the ubiquitin system. Conversely, when the Hippo pathway is inactive, YAP and TAZ shuttle to the nucleus where they bind to TEA domain (TEAD) transcription factors and regulates the transcription of many downstream genes [89]. *Taz* knockout mice show abnormal alveolarization, leading to airway enlargement that mimics human emphysema [90,91]. Conditional *Yap* knockout in lung epithelium causes disruption of bronchial morphogenesis [91]. During alveologenesis, YAP/TAZ inactivation by cell crowding orients NKX2.1 activity towards AT2 cells, a mechanism thought to be a negative feedback to limit AT1 cell expansion [92]. In bronchia, YAP is activated in distal airways, and its induction prevents multi-ciliated cell differentiation [93]. These observations suggest a general role of YAP/TAZ in favoring the stem cell compartment at the expenses of terminal differentiation. In line, YAP overexpression in adult tracheal cells results in basal cell hyperplasia and stratification [94]. Interestingly, injury in airway epithelial cells leads to downregulation of hippo signaling that increases the concentration of nuclear YAP [38], and induces FGF10 secretion by the adjacent mesenchymal cells [95]. Moreover, in the absence of *Yap*, epithelial progenitors cannot respond to local TGF-β signaling [93]. Overall, the Hippo pathway plays a critical role in lung development and response to injury directly in epithelial cells or indirectly through epithelium/mesenchyme signaling.

Other factors also are involved in lung development, such as components of the vascular endothelial growth factor (VEGF) and WNT signaling pathways. Their expression is highly regulated in space and time, allowing optimal lung development, at least partly through epithelium-mesenchyme interactions. Single-cell transcriptomic analyses will help to identify all the factors involved in lung development.

### 3.2. Extracellular Matrix Compounds

The ECM exerts functions of support, structure and stabilization that are essential for organ development and homeostasis. By directly binding to growth factors, it can also modulate the activity of secreted factors [96]. At the end of the human pseudo-glandular stage, proteoglycans, such as decorin and lumican, are located between the epithelium and the mesenchyme compartments, along with collagen type I, III and VI [97]. In ex-vivo cultures of mouse embryonic lungs, decorin binds to and neutralizes exogenous TGF-β with high affinity, thus inhibiting TGF-β signaling [98]. Furthermore, heparan sulfates present in the EMC can stabilize the interaction between growth factors and receptors. For example, Izvolsky et al. [99] showed that endogenous gradients of heparan sulfates, especially highly sulfated heparan sulfates, help lung budding induced by FGF10.

## 4. Physiological Roles of Mesenchymal Cells in Bronchioles

In the airways, epithelial cells are directly exposed to the outside environment and form a protective barrier against pathogens and toxic particles. Some bronchial epithelial cell populations contribute to epithelium repair, such as basal cells [100], one of the main airway stem cell sources [101], and club cells that can trans-differentiate for epithelium renewal [102,103,104]. Mesenchymal cells also participate by supporting lung epithelium homeostasis and repair [105,106].

Bronchial epithelium maintains the quiescence of mesenchymal cells through the Hedgehog pathway paracrine signaling [105]. Upon epithelial injury, SHH is downregulated to promote proliferation of peribronchial mesenchymal cells and the repair of the damaged tissue [105]. Peng et al. [72] showed that loss of SHH expression in airway epithelial cells leads to expansion of the surrounding mesenchymal Gli1 population that promotes the increase of the total club cell number and consequently bronchial hyperplasia.

Recently, several new lung mesenchyme cell types that can self-renew and contribute to mesenchyme self-maintenance have been described. Zepp et al. [107] characterized Axin2-Pdgfr+ mesenchymal cells that promote the regeneration and growth of alveolar cells, and Axin2+ myofibrogenic progenitors that contribute to the pathological myofibroblast response after lung injury. The origin of myofibroblasts in the lung is still debated because it requires multiple signals from fibroblasts, pericytes and other cell types [108,109]. Although it is still difficult to distinguish the different mesenchymal cell types, presumably because of an underlying continuum of differentiation, new tools such as single-cell RNA-sequencing will provide new data to expand the taxonomy of mesenchymal cells (Figure 4). In an analysis of mesenchymal cells from multiple tissues, Buechler et al. identified a dermatopontin-positive cell fibroblast population, possibly representing a universal fibroblast cell population that can give rise to other fibroblast subsets in the different tissues [110].

More recently, Fang et al. [106] tagged with a green fluorescent protein the basic helix-loop-helix transcription factor TWIST2/Dermo1 that is strongly expressed in mouse mesodermal tissues [111]. They found that after lung injury using lipopolysaccharide and naphthalene (a polycyclic aromatic hydrocarbon), the Dermo+ mesenchymal population differentiated into club cells, ciliated cells, goblet cells and neuroendocrine cells. In mice, exposure to naphthalene specifically depletes club cells by binding to CYP2F2 enzymes. These results suggest that besides their pleiotropic (e.g., anti-inflammatory and trophic) effects [112], mesenchymal cells may also directly contribute to epithelium regeneration.

## 5. Mesenchymal Cells in Chronic Obstructive Pulmonary Disease

Increased environmental susceptibility, as observed in asthma, might be seen as a failure of the innate immune system to prevent the adaptive immunity engagement and the subsequent airway inflammation. Ultimately, this might lead to cell loss and alterations that cause respiratory functional disruption. The lung physiological defense systems and regeneration capacities could be overwhelmed. In some individuals, this can lead to chronic lung diseases, such as COPD, pulmonary fibrosis and lung cancer (see Wolters et al. and Gohy et al. [113,114]). This review will focus on the role of mesenchymal cells in COPD.

### 5.1. Genetic Contribution

COPD is the third leading cause of death worldwide: more than 3.2 million of deaths in 2017 that should increase to more than 4.4 million by 2040 [115,115]. This chronic airway disease is associated with inflammation and structural changes, leading to permanent bronchial obstruction [116,117,118]. The insidious progression of the disease might explain its frequent underdiagnosis and late diagnosis [119]. Moreover, COPD is associated with many comorbidities, such as cardiac, gastrointestinal, cerebral and muscular diseases [120,121,122,123,124,125]. Tobacco smoking is the first cause of COPD in western countries [126,127,128]. Air pollution, including by biomass combustion [129,130,131], occupational and non-occupational exposure to dust and chemical agents [132], and repeated airway infections during childhood [133] are now recognized as contributing causes of COPD. Genetic and environmental factors could influence the susceptibility to COPD. Multiple large-cohort genome-wide association studies to understand the link between loci associated with lung function impairment and COPD found many polymorphisms near the Hedgehog Interacting Protein (*HHIP*) gene [134,135,136,137] that encodes a physiological inhibitor of SHH [138]. Furthermore, several studies have highlighted a relationship between susceptibility to COPD, *FGF*10 gene variants, and human airway branching variations. For instance, Smith et al., using computed tomography observed greater central airway bifurcation density, bronchial anatomic variations and narrower airway lumens in all lobes of patients with COPD than in controls. They found that these changes were significatively associated with *FGF*10 variants [139]. Gene mutations in components of key signaling pathways involved in lung development may promote COPD development.

The huge heterogeneity in COPD triggers and clinical expression may be explained by the different underlying mechanisms. Indeed, COPD should not be seen as a unique entity, but as a syndrome [140].

### 5.2. The Epithelial–Mesenchymal Crosstalk

The release of inflammatory mediators, such as TNF-α, interleukin (IL)-6 and IL-8, promotes chronic airway inflammation and ECM deposition [140,141]. In the epithelial cell compartment, goblet and basal cell hyperplasia, squamous metaplasia, mucus hypersecretion and altered cilia beating are the classical structural changes observed in patients with COPD and in smokers [4,142,143,144]. The contribution of the bronchial mesenchymal compartment to COPD progression is strongly suggested by its role in bronchial injury repair, but this is still debated. Many studies have identified an increase in the expression of mesenchymal markers, such as vimentin and the fibroblast protein S100A4, in COPD lung samples [145]. Myofibroblasts and fibroblasts could remodel the ECM by releasing matrix metalloproteinases, such as MMP9 [146,147]. Interestingly, a recent study identified different fibroblast subtypes that are localized in the lung subepithelial, subpleural and parenchymal regions and that contribute to ECM expansion in pulmonary fibrosis [24]. Similar injury-response mechanisms could be involved in COPD. These data suggest that the epithelial-mesenchymal crosstalk or the “epithelial-mesenchymal trophic unit” plays crucial roles in driving lung pathology [148].

### 5.3. Peribronchiolar Fibrosis

In COPD, the major site of obstruction is located in the small conducting airways (i.e., bronchioles with a diameter < 2 mm) [149,150]. During COPD, small airways become narrower due to the airway wall thickening and peribronchiolar fibrosis. The mechanisms of peribronchiolar fibrosis are poorly understood, and small airway fibroblasts have not been well characterized. Senescent fibroblasts could have a role in small airway fibrosis [151] due to their increased secretion of collagen 1A1 and 3A1 and increased expression of matrix metalloproteinases (MMP2, MMP9).

Cigarette smoke and oxidative stress may stimulate the release of profibrotic mediators, such as TGF-β and IL-1 beta, by airway epithelial cells [152]. Increased TGF-β1 expression (mRNA and protein) has been observed in epithelial and endothelial cells from small airways of patients with COPD compared with controls [153]. Air–liquid interface culture of airway epithelial cells from patients with COPD showed increased EMT and increased release of TGF-β that were correlated with the degree of peribronchiolar fibrosis and airway obstruction [154]. These profibrotic growth factors may induce a profibrotic phenotype in adjacent airway fibroblasts or promote the differentiation of bronchial smooth muscle cells into myofibroblasts. Indeed, a recent study showed an increase of the αSMA+ myofibroblast population in small airways of patients with COPD compared with controls [155].

### 5.4. Extracellular Matrix Composition

In COPD, ECM is degraded by enzymes, such as neutrophil elastase, metalloproteinases, hyaluronidases and chondroitinases. It has been shown that in different lung compartments, elastic fibers, elastin, glycosaminoglycans (e.g., hyaluronic acid) and type I collagen decrease drastically, but not fibronectin, tenascin and other collagens. These changes contribute to peri-bronchial fibrosis and progressive emphysema that profoundly impact the respiratory functions (Figure 5) [156,157,158,159,160]. 

### 5.5. Epithelial–Mesenchymal Transition

Cigarette smoke also induces EMT in bronchial epithelial cells of smokers and patients with COPD, thus altering epithelial function and contributing to defective lung remodeling [161]. EMT is a morphogenetic cell conversion program of epithelial cells to mesenchymal cells. EMT and mesenchymal–epithelial transition are key processes during embryogenesis and organogenesis [31]. Normally, epithelial cells are polarized and attached to the basement membrane via their basal surface. Depending on the microenvironment conditions and the cell physiological state [162], these cells can lose many epithelial characteristics, such as cell–cell adhesion (Figure 5B) and cell polarity, and acquire features of mesenchymal cells, such as migration and invasion [163,164,165]. EMT illustrates the epithelial cell phenotype plasticity. Of note, an intermediate “metastable” state where cells co-express markers of both epithelial and mesenchymal cells has been described [166,167]. In chronic airway diseases, EMT is mainly promoted through TGF-β/SMAD signaling [168] that with other signaling pathways promotes myofibroblast proliferation [169,170] and activates several ECM components. By acquiring mesenchymal cell properties, such as migration through the basement membrane, these cells could contribute to peribronchial fibrosis [145,171]. In COPD, EMT could also be promoted by the bronchial epithelium basement membrane fragmentation that facilitates pathogen penetration in the subepithelium and increases local inflammation [172,173]. Hence, mesenchymal cell proliferation and EMT might contribute to COPD pathophysiology and explain some of the observed architectural changes, such as bronchial lumen thickening (Figure 5).

### 5.6. Airway Smooth Muscle

ASM cells are the main pharmacological target in COPD through inhaled drugs, such as long-acting β-agonists, anticholinergic and corticosteroids. However, little is known about ASM cell changes in COPD. Differently from asthma, no ASM cell alteration (morphology and size) has been observed in the large airways of patients with COPD, and the ASM cell number does not correlate with airflow limitation [174]. Similarly, the proliferation rate of cultured ASM cells from patients with COPD is not increased [175]. Conversely, in small airways, ASM mass is significantly increased in COPD and is inversely correlated with lung function [176].

Some mesenchymal cell populations, such as mesenchymal stromal cells, could have a therapeutic potential in COPD. Indeed, administration of mesenchymal stromal cells in mouse [177] and in rat [178] models of COPD attenuates lung damage. However, it does not improve COPD outcomes [179,180] and therefore, its use is still debated [181,182,183]. Recently, in patients with COVID-19-associated pneumonia, the systemic delivery of allogeneic mesenchymal stromal cells (MSC) has been proposed. MSC anti-inflammatory properties and their ECM remodeling capacity suggest that MSC- based therapy may prevent fibrosis. Several clinical trials have shown the feasibility of this approach but preliminary data are mixed, possibly due to bronchial inflammation which may dampen their therapeutic efficacy [184]. Therefore, randomized trials are needed before drawing definitive conclusions [185].

## 6. Conclusions

Lung mesenchyme plays an important role in lung development and homeostasis, and an imbalance or a defect in the mesenchymal cell response to chronic injury could contribute to lung diseases. However, the exact role of the mesenchymal compartment in lung homeostasis and disease is still largely unknown. As a first step to improve this knowledge, it is essential to precisely determine the taxonomy of lung mesenchymal stromal cells during lung development and disease (a task accelerated by single-cell RNA sequencing) and the composition of that niche.

## Figures and Tables

**Figure 1 cells-10-03467-f001:**
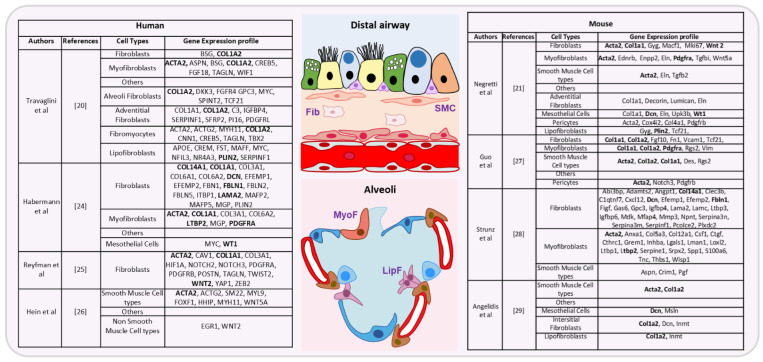
Summary of genes expressed in the different lung mesenchymal cell types (distal airways and alveoli), based on different recent studies in human (**left**) [20,24,25,26] and mouse (**right**) [21,27,28,29] samples. For fibroblasts cells (Fib), smooth muscle cell types (SMC), myofibroblasts (MyoF) and lipofibroblasts (LipF), genes present in at least two different studies are listed in bold. Distal airway: goblets, ciliated, club and basal cells are green, yellow, blue and lila, respectively. Alveoli: alveolar type 1, alveolar type 2 and pericytes are dark blue, light blue and brown, respectively.

**Figure 2 cells-10-03467-f002:**
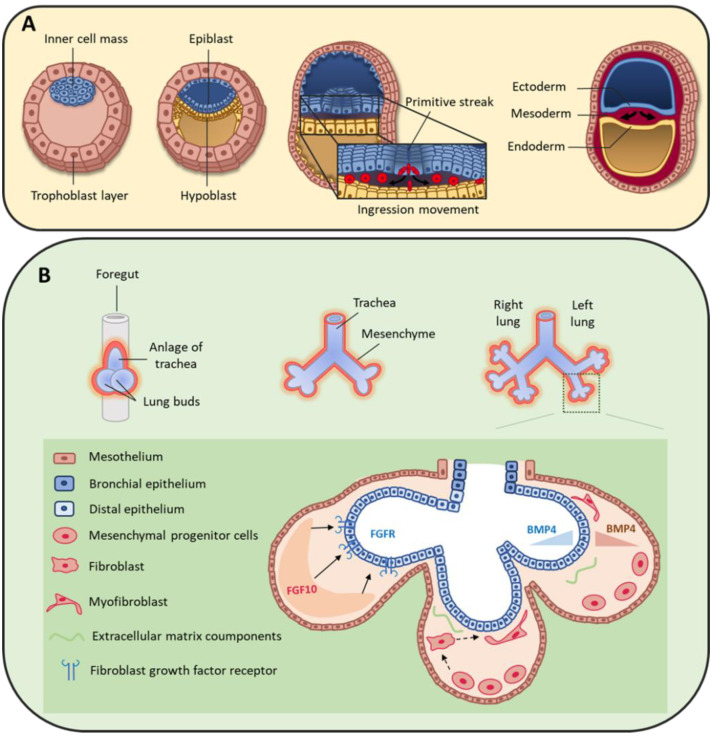
(**A**) During early human embryo development, the inner cell mass differentiates and becomes organized in the epiblast and hypoblast layers. Primitive streak formation leads to an ingression movement of epiblastic cells that elongate and detach from each other via a critical epithelial–mesenchymal transition [31]. This results in the formation of the three germ layers: ectoderm, mesoderm and endoderm. (**B**) On day 22, the foregut forms a ventral outgrowth leading to the formation of larynx and trachea in its proximal part, and lung buds in the distal part. Bifurcation and splitting of the lung buds give rise to the future right and left lungs. These structures grow ventrally to caudally through the surrounding mesenchyme. Mesenchymal progenitor cells secrete many factors, including fibroblast growth factor 10 (FGF10) that interacts with fibroblast growth factor receptor (FGFR) expressed by distal epithelial cells. Moreover, some cytokines, such as bone morphogenic protein 4 (BMP4), are secreted by both epithelial and mesenchymal cells.

**Figure 3 cells-10-03467-f003:**
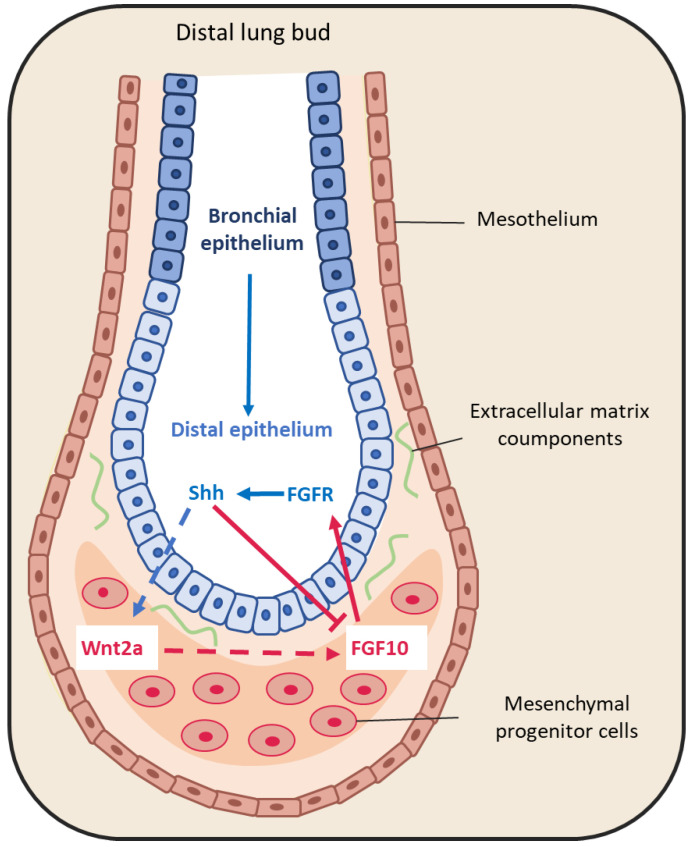
Representative schema of epithelium-mesenchyme interactions during lung budding. Many factors are secreted and exchanged between the lung epithelium (blue arrows) and mesenchyme (red arrows). For instance, fibroblast growth factor 10 (FGF10) is locally expressed by distal submesothelial mesenchymal cells and interacts with its receptor expressed in the distal epithelium. Sonic hedgehog (SHH) is expressed by epithelial cells, and downregulates FGF10 expression through its receptor Patched 1, but also activates FGF10 through the Wnt pathway.

**Figure 4 cells-10-03467-f004:**
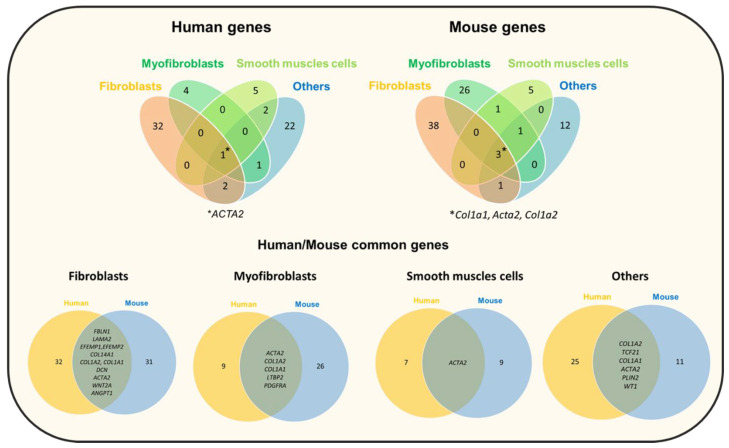
Venn diagrams showing the number of differentially expressed genes in each comparison (between human and mouse) and the overlaps between the four main compared mesenchymal cell types (fibroblasts, myofibroblasts, smooth muscle cells and other cell types). 1* and 3*, number of genes shared by the four cell types. This analysis is based on data from eight single-cell RNA sequencing studies (see Figure 1). The Venn diagrams were generated using the Venny tool.

**Figure 5 cells-10-03467-f005:**
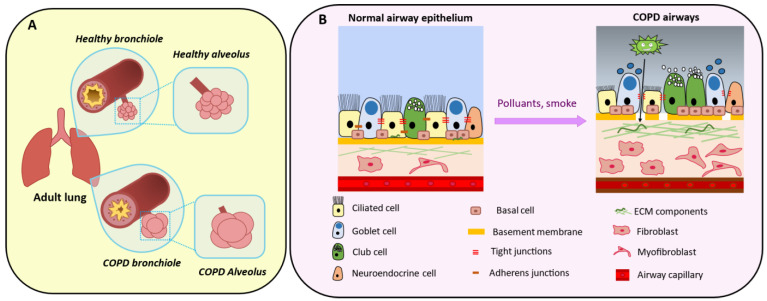
Aspects of COPD development. (**A**) (**Top panel**) Healthy bronchioles present a thin wall and a lumen where mucus is secreted to externalize pathogens. Alveoli contain elastin fibers and regulate gas exchanges between lung and blood. (**Bottom panel**) Over time, chronic exposure to tobacco, air pollution and/or recurrent infections lead to a thickened bronchiole wall associated with airflow obstruction. Emphysema is caused by gas trapping that damages the alveoli, followed by progressive peribronchiolar fibrosis. (**B**) (**Right panel**) In the alveolar-capillary unit, epithelial cells are inter-connected and attached to the basement membrane above the mesenchymal compartment. (**Left panel**) Chronic exposure to pollutants damages ciliated cells (leading to impaired mucociliary clearance), alters the epithelial cell barrier and the basement membrane with loss of tight junctions and pathogen penetration in the lower layers. This is associated with mucus plugging and goblet cell hypersecretion. Pro-proliferative fibroblasts and myofibroblasts contribute to extracellular matrix (ECM) deposition that impairs injury repair.

## Data Availability

Not applicable.

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
