# Peer review of "Roles of Mesenchymal Cells in the Lung: From Lung Development to Chronic Obstructive Pulmonary Disease"

_cells, 2021, doi:10.3390/cells10123467_

Round 1
Reviewer 1 Report
The authors carry out a review on the involvement of mesenchymal cells in lung embryonic development and in the triggering of COPD. The manuscript is easy to understand and its reading is very pleasant. The authors correctly link the various aspects addressed in this review and the mechanisms and molecules involved in the role played by mesenchymal cells in these processes.
- As an improvement to the article, I propose that the authors add a figure next to Table 1, indicating the genes expressed in the different mesenchymal cell types and that coincide in the selected studies. It can be through a Venn diagram or any other figure that they consider appropriate. This would help to check both similarities and heterogeneity.
- On page 12, line 430, the authors refer to Figure 3. Perhaps this is a typographical error and they want to refer to Figure 4.
- Some bibliographic citations are not properly referenced. For example, they should check 21,110, 142, 182 and 184.
In general terms the work is worthy of publication.
Author Response
The authors carry out a review on the involvement of mesenchymal cells in lung embryonic development and in the triggering of COPD. The manuscript is easy to understand and its reading is very pleasant. The authors correctly link the various aspects addressed in this review and the mechanisms and molecules involved in the role played by mesenchymal cells in these processes.
As an improvement to the article, I propose that the authors add a figure next to Table 1, indicating the genes expressed in the different mesenchymal cell types and that coincide in the selected studies. It can be through a Venn diagram or any other figure that they consider appropriate. This would help to check both similarities and heterogeneity.
* We thank the reviewer for this insightful suggestion. In this revised document, we modified Table 1 that is now Figure 1 and that includes both former table 1, with genes from the same cell type present in at least two different studies listed in bold, and two figures indicating the location of the main cell types in distal airways and alveoli.
On page 12, line 430, the authors refer to Figure 3. Perhaps this is a typographical error and they want to refer to Figure 4.
* Indeed, this was a typographical error that was corrected.
Some bibliographic citations are not properly referenced. For example, they should check 21,110, 142, 182 and 184.
* We thank the reviewer to have identified these errors, that were corrected.
In general terms the work is worthy of publication.
* We thank the reviewer for this kind comment on our review.
Reviewer 2 Report
Amel Nasri et al., described the complicated roles of mesenchymal cells from lung development to pathological characteristics. This review article summarized the gene expression profiles of different types of lung mesenchymal cells which will enable researchers to dissect specific types of mesenchymal cells in an easier manner. Next, the authors introduced how mesenchymal cells regulate lung development via the crosstalk with other cells such as fibroblast, myofibroblast, epithelial, and epithelium cells. Besides, the growth factors and cytokines were also taken into consideration with mesenchymal cells in lung organogenesis. Next, the authors summarized the physiological roles of mesenchymal cells in bronchioles. The mesenchymal cells can self-renew and contribute to mesenchyme self-maintenance usually for repairing in the injured condition. The authors also described that Axin2+ myofibrogenic progenitors contribute to the pathological myofibroblast response after lung injury, shaping the possible route in diminishing pathological effects of mesenchymal cells. Next, the authors summarized the epithelial-mesenchymal crosstalk via various cytokines and secreted factors such as TNF-α, interleukin (IL)-6, and IL-8 which were known to promote chronic airway inflammation and ECM deposition. The authors also indicated that the abnormal depositions of ECM mediated by mesenchymal cells are major routes to induce peribronchiolar fibrosis leading to airway wall thickening and narrowed small airways during COPD, implying a possible direction to attenuate COPD. Overall, this review article is well-conducted, the logics are properly and the dissection of lung mesenchymal cells in development, physiological and pathological condition is thorough.
Author Response
We thank the reviewer for this kind comment on our review.
Reviewer 3 Report
In this review, authors updated recent studies on crosstalk between epithelium and mesenchyme during lung development and diseases with COPD. The manuscript is well written and informative to the field.
Author Response
We thank the reviewer for this positive feedback.